# Influence of GST- and P450-based metabolic resistance to pyrethroids on blood feeding in the major African malaria vector *Anopheles funestus*

Lynda Nouage[1,2]*, Emmanuel Elanga-Ndille[1]*, Achille Binyang[1,2], Magellan Tchouakui[1,2], Tatiane Atsatse[1,2], Cyrille Ndo[3,4], Sévilor Kekeunou[2], Charles S. Wondji[1,5]

**1** Department of Medical Entomology, Centre for Research in Infectious Diseases (CRID), Yaoundé, Cameroon, **2** Department of Animal Biology and Physiology, Faculty of Science, University of Yaoundé 1, Yaoundé, Cameroon, **3** Department of Parasitology and Microbiology, Centre for Research in Infectious Diseases (CRID), Yaoundé, Cameroon, **4** Department of Biological Sciences, Faculty of Medicine and Pharmaceutical Sciences, University of Douala, Douala, Cameroon, **5** Department of Vector Biology, Liverpool School of Tropical Medicine, Liverpool, United Kingdom

☯ These authors contributed equally to this work.
* emmsdille@yahoo.fr, emmanuel.elanga@crid-cam.net (EEN); lnouage@gmail.com (LN)

**Data Availability Statement:** All relevant data are within the paper.

## Abstract

Insecticide resistance genes are often associated with pleiotropic effects on various mosquito life-history traits. However, very little information is available on the impact of insecticide resistance on blood feeding process in mosquitoes. Here, using two recently detected DNA-based metabolic markers in the major malaria vector, *An. funestus*, we investigated how metabolic resistance genes could affect the blood meal intake. After allowing both the field F1 and lab F8 *Anopheles funestus* strains to feed on the human arm for 30 minutes, we assessed the association between key parameters of blood meal process including, probing time, feeding duration, blood feeding success, blood meal size, and markers of glutathione S-transferase (*L119F-GSTe2*) and cytochrome P450 (*CYP6P9a_R*)—mediated metabolic resistance. None of the parameters of blood meal process was associated with *L119F-GSTe2* genotypes. By contrast, for *CYP6P9a*_R, homozygous resistant mosquitoes were significantly more able to blood-feed than homozygous susceptible (OR = 3.3; CI 95%: 1.4–7.7; P = 0.01) mosquitoes. Moreover, the volume of blood meal ingested by CYP6P9a-SS mosquitoes was lower than that of CYP6P9a-RS (P<0.004) and of CYP6P9a-RR (P<0.006). This suggests that *CYP6P9a* gene is inked with the feeding success and blood meal size of *An. funestus*. However, no correlation was found in the expression of *CYP6P9a* and that of genes encoding for salivary proteins involved in blood meal process. This study suggests that P450-based metabolic resistance may influence the blood feeding process of *Anopheles funestus* mosquito and consequently its ability to transmit malaria parasites.

**Funding:** This study was funded by a Wellcome Trust Training fellowship (109930/Z/15/Z) awarded to EEN.

**Competing interests:** The authors have declared that no competing interests exist.

## Introduction

Malaria remains a major public health scourge in sub-Sahara Africa despite significant progress made since the 2000s in reducing its burden [1]. This disease is caused by a *Plasmodium* parasite transmitted by *Anopheles* mosquito species while taking a blood meal from humans. Blood feeding is essential for female mosquito's fecundity [2] as *Anopheles* species like all anautogenous female mosquitoes, require a blood meal to obtain amino acids needed to synthesize yolk proteins for eggs maturation [3, 4]. The blood feeding success of mosquitoes is facilitated by the biochemical proprieties of salivary gland proteins [5]. Indeed, some salivary proteins such as anopheline antiplatelet protein (AAPP), apyrase, gambiae Salivary Gland protein 6 (gSG6) and members of D7 family have been identified as vasodilators, anti-coagulants and inhibitors of platelet aggregation allowing mosquitoes to overcome host haemostatic mechanisms and to have a successful blood meal [5–8]. Mosquito fecundity was shown to vary by source and size of the blood meal with a difference of these two parameters resulting in significant variations of the number of eggs laid by each female mosquito [4, 8]. It has been shown that the number of eggs laid per female is positively associated to the amount of blood ingested as larger blood meals resulted to an increase of the number of females that developed eggs and the number of eggs per female [9, 10]. The volume of blood taken by a mosquito could be affected by a range of intrinsic (host immunity) and extrinsic factors including ambient temperatures, mosquito age, parity status, gonotrophic cycle, blood feeding history and infection status [10]. More recently, it was reported that exposure to pyrethroids could also significantly influence the process of taking a blood meal and the blood meal volume ingested by Kdr-resistant *Anopheles gambiae* females [11]. Pyrethroids (PY) are the insecticide class mostly used in the last two decades through ITNs and IRS strategies to control malaria transmission [12]. Unfortunately, the widespread use of these insecticides has favoured the development of resistance in malaria vector species [13, 14]. Resistance to pyrethroids involves two main mechanisms: (i) metabolic resistance, due to the increase expression level of detoxifying enzymes, belonging to three families: the cytochrome P450 monooxygenases, the glutathione S-transferases and the carboxylesterases; and (ii) target-site resistance due to mutations in the voltage sodium channels known as knock-down (kdr) mutations [15, 16]. Although resistance mechanisms help mosquitoes to survive under continuous insecticide pressure, these actions are costly and may negatively affect mosquito's fitness including body size, adult longevity, larval development time, fecundity, fertility, mating competitiveness and blood feeding capability [17–19]. For target-site resistance, a decreased longevity and an increased larval development time have been reported in *kdr*-pyrethroid-resistant mosquitoes [20, 21]. Moreover, a recent study suggested that kdr-based resistance could impact blood feeding with heterozygote (*kdr*-RS) and susceptible (*kdr*-SS) mosquitoes taking higher blood volume than homozygote (*kdr*-RR) resistant individuals [11]. In some cases, resistant mosquitoes displayed a significant advantage compared to their susceptible counterparts as shown recently for female longevity [22] and vectorial capacity [23]. However, little is known on the impact of metabolic resistance as DNA-based markers were not previously available for this mechanism; thereby limiting the ability to investigate its physiological impact on the blood feeding process in mosquitoes. However, taking advantage of the identification of the first DNA-based metabolic marker in *An. funestus* mosquito, one study reported that a GST-based metabolic resistance caused by a leucine to phenylalanine amino acid change at codon 119 in the glutathione S-transferase epsilon 2 *(L119F-GSTe2)* [24], has a detrimental impact on *An. funestus* fitness. The authors reported that field-resistant mosquitoes exhibited a reduced fecundity and slower larval development but an increased adult longevity [22]. On the other hand, a new DNA-based assay was recently designed for cytochrome P450-mediated resistance (the *CYP6P9a*-R) in *An. funestus*. This

marker showed that mosquitoes carrying this P450-resistant allele survived and succeeded in blood feeding more often than did susceptible mosquitoes when exposed to insecticide-treated nets [25]. The design of assays for both GST- and P450-based resistance now offers a great opportunity to explore how the blood feeding process is influenced by metabolic resistance mechanisms in malaria vectors and further assess how resistance may impact the vectorial capacity of mosquitoes to transmit malaria in the natural environment.

Here, we investigated the effect of metabolic resistance to pyrethroids on the blood feeding process in *An. funestus*, using the two DNA-based metabolic resistance markers: *L119F-GSTe2* and *CYP6P9a*-R [24, 25]. Specifically, we assessed the association between the genotypes of these metabolic resistance markers and key parameters of blood feeding including mosquito probing time, feeding duration and the blood meal size.

## Material and methods

### Mosquito collection and rearing

Experiments were carried out using both field and lab strains of *An. funestus*. Field mosquitoes ($F_1$) were generated from indoor resting female (F0) collected in Mibellon (6˚46'N, $11^0$ 70'E), a village located in a rural area of the savanna-forest region in Cameroon, Central Africa where the *L119F-GSTe2* has been reported [26]. Blood-fed field collected females were kept in paper cups and transported to the insectary of the Centre for Research in Infectious Diseases (CRID) in Yaoundé where they were kept for 4–5 days until they became fully gravid and were then induced to lay eggs using the forced eggs-laying method [27]. The eggs were placed in paper cups containing water to hatch, after which the larvae were transferred into trays and reared to adults. To assess the effect of *CYP6P9a* marker, $F_8$ progenies were generated from crosses established between the pyrethroid susceptible laboratory strain (FANG) and the resistant (FUMOZ-R) lab strain. These two *An. funestus* lab strains were colonized from mosquitoes collected in Southern Africa region. FUMOZ is a pyrethroid resistant strain established in the insectary from wild-caught *An. funestus* mosquito species from southern Mozambique [28]. The previous study reported that the over-expression of two duplicated P450 genes, *CYP6P9a* and *CYP6P9b*, constitute the main mechanism driving pyrethroid resistance in this strain [29, 30] for which the *119F-GSTe2* allele is absent [24]. The FANG strain is completely susceptible to pyrethroids colonized from Calueque in southern Angola [28].

### Blood feeding experiments and blood meal size quantification

**Blood feeding process.**   Since blood meal volume has previously been reported to correlate with mosquito size [2], individuals used for blood feeding experiments were firstly starved for 24h then grouped according to their size. Mosquito size was determined by weighing (using an analytical micro-scale, SARTORIUS, Goettingen, Germany). Each starved individual (adult females aged 3–7 days) was immobilized by chilling for 2 minutes at 5˚C. Each mosquito was then placed in paper cups covered with black sheet for about an hour before given a blood meal. In order to evaluate the association between metabolic resistant markers on blood meal size, mosquitoes were allowed to bite for 30 min on the bare forearm of a single human volunteer and then genotyped for L119F-GSTe2 and CYP6P9a. Ethical clearance was obtained from the National Ethics Committee of Cameroon's Ministry of Public Health (N˚2018/04/1000/CE/CNERSH/SP) in conformity to the WMA Declaration of Helsinki. Informed verbal consent was obtained from household owners for using their houses for mosquito collection.

The duration of probing and blood feeding was assessed using a batch of 120 $F_1$ female field-collected mosquitoes. For this purpose, mosquitoes were individually transferred in polystyrene plastic cups covered with netting. They were allowed to rest for 15 min before

observations began. During the blood intake, each mosquito was filmed with a Digital HD Video Camera (Canon PC2154, Canon INC, Japan) placed beside the plastic cup. At the end of the time allowed for feeding, the film for each mosquito was analysed and the parameters such as probing time (defined as the time taken from initial insertion of the mouthparts in the skin until the initial engorgement of blood) [5] and total feeding duration, were recorded, using a digital timer. Due to the low density of female mosquitoes obtained at $F_8$ generation from crosses of the lab strain mosquitoes, experiments to estimate the probing and the feeding duration of this strain were not investigated.

To determine the blood meal size for both strains, batches of 25 mosquitoes grouped according to their weight were allowed to bite on a human arm. In this case, neither the probing time nor the feeding duration was recorded. After the trial, the whole abdomen of successfully fed mosquitoes (evident by red-coloration engorgement of the abdomen) was extracted and stored in an individual 1.5 ml microtube at– 20°C to measure the blood meal size. The rest of the carcasses as well as unfed mosquitoes were kept individually in a microtube containing RNA-later and stored at -20°C.

## Blood meal size quantification

To compare the volume of blood ingested between resistant and susceptible mosquitoes, the volume of blood ingested by each mosquito was determined by quantifying the haemoglobin amount, as previously described [31]. Briefly, abdomens of blood fed mosquitoes were homogenized in 0.5 ml of Drabkin's reagent (containing 1.0g of sodium bicarbonate, 0.1g potassium carbonate, 0.05g potassium cyanide, 0.2g potassium ferricyanide all diluted in 1L of distilled water). This reagents converts the haemoglobin into haemoglobin cyanide (HiCN). After 20 minutes at room temperature and the addition of 0.5 ml of chloroform solution, samples were centrifuged at 5600 rpm (3512 rcf) for 5 min. The aqueous supernatant containing HiCN was transferred in a new 1.5 ml microtube. An aliquot of 200µl from each sample was transferred to a microplate and the optical density (OD) read at a wavelength of 620nm in a spectrophotometer (EZ Read 400, biochrom, Cambridge, UK). OD for each sample were read in duplicate and the average value between the two replicates was considered as OD value of the sample. In parallel, OD read on various amounts of human volunteer blood added to Drabkin's reagent in individual microtubes were used as control to generate calibration curves and the regression line used to assess the relationship between OD and blood volume. For each sample, the blood meal size was estimated according to the weight by dividing the blood volume estimated using the regression line by the average weigh of each batch of mosquitoes constituted after the weighing. The blood meal size was then expressed in µL of blood per mg of weight.

## Molecular species identification

To determine the species composition of *An. funestus* group among the samples, genomic DNA (gDNA) was extracted from both blood-fed and unfed mosquitoes using the Livak protocol [32]. Instead of using the whole body as done for unfed mosquitoes, DNA was extracted from the carcasses of fed mosquitoes after removing the abdomen for blood volume quantification. The concentration and purity of the extracted gDNA were subsequently determined using a NanoDrop™ spectrophotometer (Thermo Scientific, Wilmington, USA) before storage at −20°C. An aliquot of gDNA extracted from field-collected strain was used for molecular identification by a polymerase chain reaction [33].

## Genotyping of *L119F-GSTe2* mutation in field-collected strain

The *L119F-GSTe2* mutation was genotyped using gDNA extracted from carcasses of field-collected strains following an allele-specific PCR diagnostic assay previously described [22]. The primers sequences are given in S1 Table. PCR was performed in Gene Touch thermal cycler (Model TC-E-48DA, Hangzhou, 310053, China), in a reaction volume of 15 μl using 10 μM of each primer, 10X Kapa Taq buffer A, 0.2 mM dNTPs, 1.5 mM MgCl2, 1U Kapa Taq (Kapa Biosystems, Wilmington, MA, USA) and 1μl of genomic DNA as template. The cycle parameters were: 1 cycle at 95 ˚C for 2 min; 30 cycles of 94 ˚C for 30 s, 58 ˚C for 30 s, 72 ˚C for 1 min and then a final extension at 72 ˚C for 10 min. The PCR products were size separated on a 2% agarose gel stained with Midori Green Advance DNA Stain (Nippon genetics Europe GmbH) and visualised using a gel imaging system. The size of the diagnostic band was 523 bp for homozygous resistant (RR) and 312 bp for homozygous susceptible (SS), while heterozygous (RS) showed the two bands.

## Genotyping of *CYP6P9a*-R allele in lab strain mosquitoes

The *CYP6P9a* resistance marker was genotyped using the protocol recently described in [25]. A PCR-RFLP were carried out using gDNA extracted from the carcasses of $F_8$ generation individuals obtained from the reciprocal crosses between FANG and FUMOZ strains used for blood feeding. Briefly, a partial *CYP6P9a* upstream region was amplified in a final volume of 15μl PCR mixture containing 10X Kapa Taq buffer A (Kapa Biosystems, Wilmington, MA, USA), 5 U/μl KAPA taq, 25μM dNTP, 25μM MgCl2, 10 mM of each primer, 10.49μl of dH2O and 1μl of genomic DNA. The PCR cycle parameters were as follows: the initial denaturation step at 95˚C for 5 minutes followed by 35 cycles of 94˚C for 30 seconds, 58˚C for 30 seconds and 72˚C for 45 seconds and a final extension step of 72˚C for 10 minutes. The PCR products were size separated on a 1.5% agarose gel stained with Midori Green Advance DNA Stain (Nippon genetics Europe GmbH) and visualised using a gel imaging system to confirm the product size (450bp). Then, the PCR product was incubated at 65˚C for 2 hours. This was done in 0.2ml PCR strip tubes using 5μl of PCR product, 1μl of cutSmart buffer, 0.2μl of 2 units of Taq1 enzyme (New England Biolabs, catalog: ER0672) and 3.8μl of dH20. Size separation was performed on a 2.0% agarose gel stained with Midori Green Advance DNA Stain at 100V for 30 minutes. The gel was visualised using an ultraviolet light transilluminator.

## Gene expression profiling of major salivary genes encoding proteins involved in blood meal process

The expression profiles of a set of salivary genes encoding for proteins involved in blood meal process was compared between CYP6P9a-RR, CYP6P9a-RS and CYP6P9a-SS *An. funestus* mosquitoes. For each gene, two pairs of exon-spanning primers was designed for each gene using Primer3 online software (v4.0.0; http://bioinfo.ut.ee/primer3/) and only primers with PCR efficiency between 90 and 110% determined using a cDNA dilution series obtained from a single sample, were used for qPCR analysis. Taking into account this criteria of efficiency, only the AAPP and four members of D7 family genes (D7r1, D7r2, D7r3, and D7r4) were used for this analysis. Primers are listed in S1 Table. Total RNA was extracted from three batches of 10 whole females of 3–5 days old from CYP6P9a-RR, CYP6P9a-RS and CYP6P9a-SS mosquitoes. RNA was isolated using the RNAeasy Mini kit (Qiagen) according to the manufacturer's instructions. The RNA quantity was assessed using a NanoDrop ND1000 spectrophotometer (Thermo Fisher) and 1μg from each of the three biological replicates for each batch of mosquitoes was used as a template for cDNA synthesis using the SuperScript III (Invitrogen,

Waltham, Massachusetts, USA) with oligo-dT20 and RNase H, following the manufacturer's instructions. The qPCR assays were carried out in a MX 3005 real-time PCR system (Agilent, Santa Clara, CA 95051, United States) using Brilliant III Ultra-Fast SYBR Green qPCR Master Mix (Agilent). A total of 10 ng of cDNA from each sample was used as template in a three-step program involving a denaturation at 95 °C for 3 min followed by 40 cycles of 10 s at 95 °C and 10 s at 60 °C and a last step of 1 min at 95 °C, 30 s at 55 °C, and 30 s at 95 °C. The relative expression and fold-change of each target gene in CYP6P9a-RR and CYP6P9a-RS relative to CYP6P9a-SS was calculated according to the $2^{-\Delta\Delta CT}$ method incorporating PCR efficiency after normalization with the housekeeping RSP7 ribosomal protein S7 (VectorBase ID: AFUN007153) and the actin 5C (vectorBase ID: AFUN006819) genes for *An. funestus*.

## Statistical analysis

All analyses were conducted using GraphPad Prism version 7.00 software (GraphPad Prism version 7.00 for Windows, GraphPad Software, La Jolla California USA, www.graphpad. com"). We estimated a Fisher's exact probability test and the odds-ratio of *L119F-GSTe2* and *CYP6P9a* genotypes (homozygous resistant = RR, heterozygote resistant = RS and homozygous resistant = SS) and both susceptible (S) and resistant (R) alleles. This allowed to the assessment of the association between a) insecticide resistance and mosquito's weight by comparing the proportions of the genotypes of both genes in each group established after weighing; b) blood feeding success and insecticide resistance by comparing the proportion of each genotype in both fed and unfed mosquitoes. The feeding duration was grouped into four intervals with 1 minute (60s) amplitude for each interval. The duration of probing and feeding was analysed by comparing the proportion of mosquitoes (for *L119F-GSTe2* and *CYP6P9a*) with different genotypes in each defined intervals using chi-square test. After estimating the median of weighted blood meal for each genotype, Kruskal-Wallis and Mann-Whitney tests were used to compare the differences between more than two groups and between two groups, respectively as Shapiro-Wilks test showed non-normal distribution.

## Results

### Metabolic resistance genes and *An. funestus* mosquito's weight

A total of 1,200 ($F_1$ generation) field strains and 273 ($F_8$ generation) female mosquitoes were weighed. The mean weight of a mosquito was 0.9 ± 0.010 mg (minimum = 0.2 mg; maximum = 2.3mg) and 0.89 ± 0.016 mg (minimum = 0.2 mg; maximum = 1.7mg) for field and lab strain respectively. No significant difference was found between the mean weights of two strains ($p = 0.18$). For all the analyses, we arbitrarily grouped mosquitoes according to their weight values, into two different classes as followed: [0–1.0] mg and [1.1–2.3] mg. Analysis of the distribution of *L119F-GSTe2* mutation genotypes in each class of field strain mosquitoes showed no association between the mosquito's weight and *L119F-GSTe2* genotypes ($\chi^2 = 0.15$; $p = 0.9$; OR = 1.2, 95%CI: 0.3742–4.176, for RR *vs* RS; OR = 1.1, 95%CI: 0.3659–3.606 for RR *vs* SS; OR = 0.9, 95%CI: 0.4943–1.709 for RS *vs* SS) (Fig 1 and Table 1). This absence of correlation between the *L119F-GSTe2* genotypes and the weight of mosquitoes was confirmed at the allele level (OR = 1; 95%: CI: 0.5–2.0; $p = 0.5$) showing that the L119F mutation may not associated with the weight of this *An. funestus* field population (Table 1). In contrast, a significant association was observed between *CYP6P9a* genotypes and the weight of mosquito ($\chi^2 = 29.54$, p<0.0001). Indeed, proportions of RR and RS genotypes were higher than that of SS in the lowest weight class, whereas, for larger weight, mosquitoes with SS genotype were more abundant (67.2%). This association is further supported by odds ratio estimates showing that proportions of homozygote resistant (RR) (OR = 5.4; CI 95%: 2.3–12.7; $p$<0.0001) and

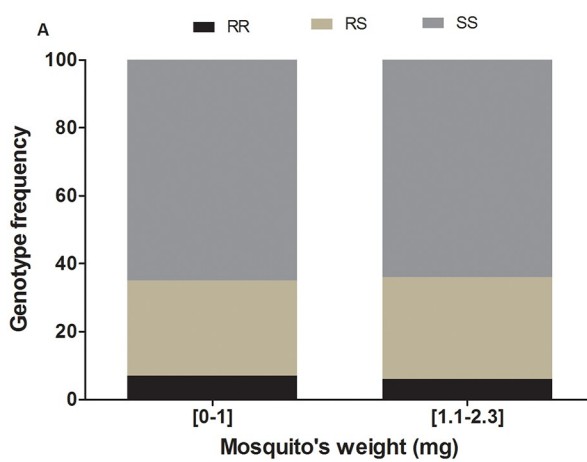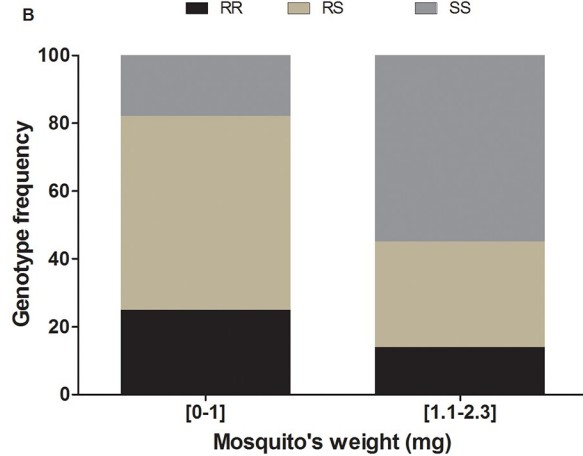

**Fig 1. Effect of metabolic resistance on *An. funestus* mosquito weight.** Distribution of genotypes of *L119F-GSTe2* (A) and *CYP6P9a*-R (B) markers according to the weight.

heterozygote (RS) (OR = 5.6; CI 95%: 2.8–11.1; $p<0.0001$) mosquitoes are significantly higher in lowest weight class than the larger one when compared to homozygote susceptible mosquitoes (Table 1). Overall, mosquitoes harbouring the *CYP6P9a*-S susceptible allele displayed higher weight compared to those with the *CYP6P9a*-R resistant allele (OR = 2.8; CI 95%: 1.5–5.0; p = 0.0003 (Table 1) suggesting that over-expression of the *CYP6P9a* gene is reducing the weight of pyrethroid resistant *An. funestus* mosquitoes.

### Influence of *L119F-GSTe2* and *CYP6P9a* mutations on *An. funestus* blood feeding success

**L119F-GSTe2.** Out of the 1,200 individuals from field strain mosquitoes that were allowed to take a blood meal, 457 (39.6%) successfully fed whereas 743 did not. Among blood-fed mosquitoes, a total of 360 were successfully genotyped and 7% (24/360) were homozygous resistant (RR), 28% (103/360) were heterozygous resistant (RS) and 65% (233/360) were homozygous susceptible (SS) (Fig 2a). On the other hand, out of the 300 unfed mosquitoes randomly selected for genotyping, 5% (15/300), 32% (62/300) and 63% (189/300), were homozygous resistant, heterozygotes and homozygote susceptible, respectively (Fig 2a). However, the distribution of L119F genotypes was not statistically different between blood-fed and unfed mosquitoes ($\chi^2$ = 0.63, p = 0.7). Furthermore, the estimation of odds ratio (OR = 1; CI 95%: 0.5–2.0;

**Table 1. level of association of *L119F-GSTe2* and *CYP6P9a*-R genotypes with mosquito weight by comparing low (0–1.0mg) and high (1–2.4mg) weight samples.**

| Genotypes | *L119F-GSTe2* | | *CYP6P9a*-R | |
|---|---|---|---|---|
| | Odds ratio | *p*-value | Odds ratio | *p*-value |
| **RR vs SS** | 1.1 (0.4–3.6) | 0.5 | 5.4 (2.3–12.7) | < 0.0001 |
| **RS vs SS** | 0.9 (0.5–1.7) | 0.4 | 5.6 (2.8–11.1) | < 0.0001 |
| **RR vs RS** | 1.2 (0.4–4.1) | 0.5 | 1.0 (0.5–2.3) | 0.5 |
| **S vs R** | 1 (0.5–2.0) | 0.5 | 2.8 (1.5–5.0) | 0.0003 |

SS: homozygote susceptible; RR: homozygote resistant; RS: heterozygote;

* significant difference p < 0.05.

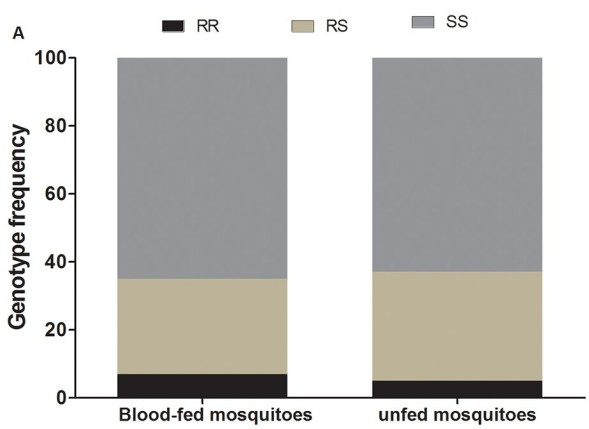
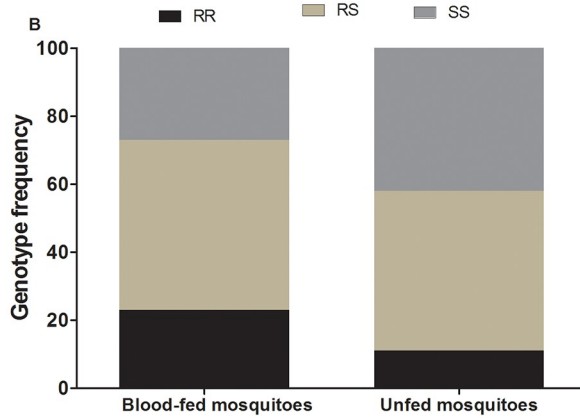

**Fig 2. Association between resistance markers and bloodfeeding.** Distribution of *L119F-GSTe2* (A) and *CYP6P9a-R* (B) genotypes between blood-fed and unfed *An. funestus* mosquitoes.

*p* = 0.6) showed overall that mosquitoes bearing the 119F-R resistant allele have the same chance to have a successful blood feeding than those with the 119F-S susceptible alleles (Table 2). This suggests that the ability to take blood is not associated with the *L119F-GSTe2* mutation in *An. funestus.*

**CYP6P9a-R.** Among a total of 273 mosquitoes that were offered a blood meal 140 successfully fed (51.3%) whereas, 133 did not. Out of the 140 mosquitoes that blood-fed, 134 were successfully genotyped for *CYP6P9a*-R allele revealing that 23% (31/134), 50% (67/134) and 27% (36/134) were homozygote resistant CYP6P9a-RR, heterozygotes CYP6P9a-RS and homozygote susceptible CYP6P9a-SS, respectively (Fig 2b). Among the unfed mosquitoes, 11.3% (15/133) were homozygote resistant CYP6P9a-RR, 47.4% (63/133) heterozygotes and 41.3% (55/133) were homozygote susceptible CYP6P9a-SS. The estimation of odds ratio showed that homozygote resistant CYP6P9a-RR mosquitoes are significantly more able to blood feed than homozygote susceptible (OR = 3.33; CI 95%: 1.4–7.7; *p* = 0.01). No difference was observed between heterozygote and homozygote resistant CYP6P9a-RR (OR = 1.9, 95%CI: 0.9–4.4; p = 0.1) neither with homozygote susceptible CYP6P9a-SS (OR = 1.7, 95%CI: 0.9–3.1; p = 0.1) mosquitoes (Table 2). Moreover, it was overall observed that mosquitoes with the *CYP6P9a*-R resistant allele have a greater chance to blood feed than those bearing the susceptible allele (OR = 1.9; CI 95%: 1.03–3.2; *p* = 0.04) (Table 2).

**Table 2. Assessment of the association of *L119F-GSTe2* and *CYP6P9a-R* mutations with *An. funestus* mosquito blood feeding.**

| | *L119F-GSTe2* | | *CYP6P9a-R* | |
|---|---|---|---|---|
| **Genotypes** | **Odds ratio** | ***p*-value** | **Odd ratio** | ***p*-value** |
| **RR vs SS** | 0.7 (0.2–2.4) | 0.4 | 3.3 (1.4–7.7) | 0.01 |
| RS vs SS | 1.1 (0.62–2.1) | 0.4 | 1.7 (0.9–3.1) | 0.1 |
| RR vs RS | 0.6 (0.2–2.3) | 0.3 | 1.9 (0.9–4.4) | 0.1 |
| R vs S | 1 (0.5–2.0) | 0.6 | 1.8 (1.1–3.2) | 0.04 |

SS: homozygote susceptible; RR: homozygote resistant; RS: heterozygote;

\* significant difference p < 0.05.

### Association between the *L119F-GSTe2* mutation and probing/ blood feeding duration

Out of the 120 mosquitoes that were individually filmed to assess the influence of insecticide resistance genes on the probing and feeding duration, 7 (6.14%), 40 (35.08%) and 67 (58.77%) were genotyped as homozygous resistant 119F/F-RR, heterozygous L119F-RS and homozygous susceptible L/L119, respectively. Overall, regardless of the genotype, the median value of mosquito's probing duration was 49.5 seconds (minimum = 4s and maximum = 290s). No difference was observed in the probing time of resistant mosquitoes 119F/F-RR (Median = 53 seconds) and heterozygotes L119F-RS (Median = 52s) compared to the homozygote susceptible L/L119 (Median = 52s).

Regarding the blood feeding duration, it was observed that the median and mean time for a mosquito to have a full blood meal was 249.5 seconds and 303 ± 181 seconds respectively, with a minimum = 68 seconds and a maximum = 772 seconds. The feeding duration was longer (median = 269s) in L/L119 mosquitoes compared to L119F-RS (229.5s) and 119F/F-RR (214s) but the difference was not statistically significant ($p$ = 0.19, Kruskal-Wallis test).

### Effect of *L119F-GSTe2* and *CYP6P9a*-R mutations on the blood meal size of *An. funestus*

**L119F-GSTe2.**   From 457 individuals that took a full blood meal it was observed that the average weighted blood meal of a mosquito regardless of the *L119F-GSTe2* genotype was 3.4± 1.3 μl/mg (minimum = 1.2 μl/mg; maximum = 9.2 μl/mg). However, the weighted blood meal was not significantly different (P = 0.17; Kruskal-Wallis test; Fig 3a) in homozygote susceptible L119-SS (3.0μl/mg) compared to homozygote resistant L119-RR (2.8μl/mg) and heterozygote L119F-RS (3.3μl/mg) mosquitoes. This result suggests that the *L119F-GSTe2* mutation is not associated with the volume of blood meal ingested by *An. funestus*.

**CYP6P9a-R.**   The influence of the *CYP6P9a*-R mutation on the volume of blood meal taken by *An. funestus*, was assessed using the 134 blood fed mosquitoes that were successfully genotyped for *CYP6P9a*-R allele. Overall, irrespective of the genotype, the mean weighted blood volume ingested by a mosquito was 4.8 ± 2 μl/mg (minimum = 2 μl/mg; maximum = 13.3μl/mg). However, the weighted blood meal volume of CYP6P9a-SS mosquitoes (Median = 3.71μl/mg) was lower than that of CYP6P9a-RS (Median = 4.73 μl/mg) and of CYP6P9a-RR (Median = 4.78 μl/mg) (Fig 3; p<0.004 for RS vs SS and p<0.006 for RR vs SS,

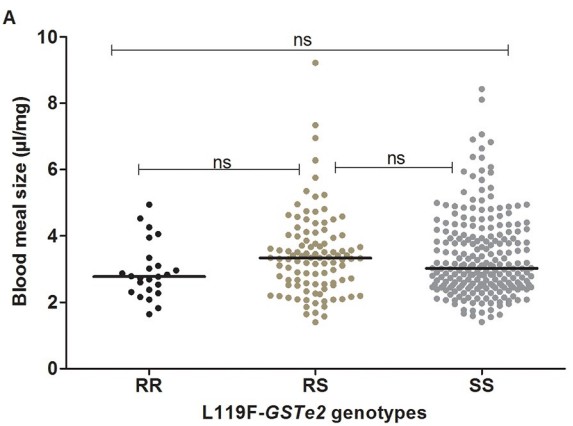
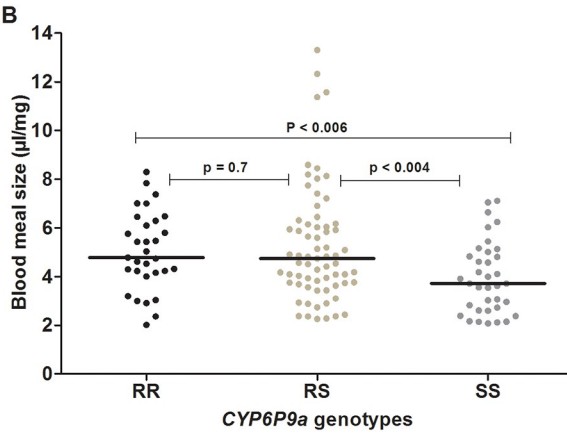

**Fig 3. Influence of metabolic resistance on blood meal size of *An. funestus* mosquitoes.** Effect of *L119F-GSTe2* (A) and *CYP6P9a*-R (B).

Mann-Whitney test). No difference in the volume of the blood meal was observed between CYP6P9a-RR and CYP6P9a-RS mosquitoes (P = 0.7; Mann-Whitney test). This result suggests that the over-expression of *CYP6P9a* gene is associated with an increase of the volume of the blood meal ingested by *An. funestus*.

### Expression profile of *AAPP* and *D7* family salivary genes according to *CYP6P9a*-R genotypes

Due to the association observed between the *CYP6P9a*-R genotypes and blood feeding, an attempt was made to assess whether the genotypes of this gene could be possible associated with the expression profile of key salivary genes. The expression level of AAPP and 4 members of the D7 family salivary genes (D7r1, D7r2, D7r3 and D7r4) was analysed and compared between homozygous resistant (CYP6P9a-RR), heterozygous (CYP6P9a-RS) and homozygous susceptible genotype (CYP6P9a-SS) mosquitoes. No significant difference in the expression level of these genes was observed between the three types of mosquitoes although D7 family genes appeared slightly over-expressed in CYP6P9a-RR and CYP6P9a-RS when compared to CYP6P9a-SS (Fig 4). This result suggests that *CYP6P9a*-R genotypes do not influence the expression profile of both AAPP and D7 family genes in the salivary glands of *An. funestus* mosquitoes.

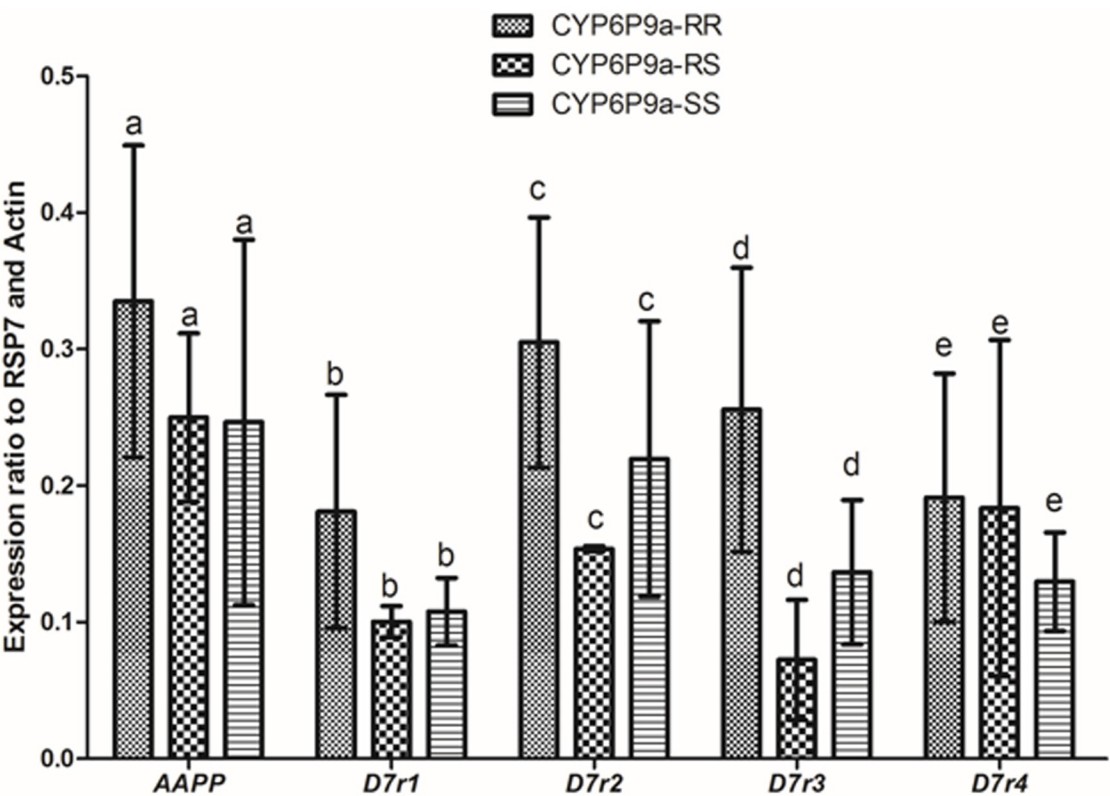

**Fig 4. Comparative expression of salivary genes between CYP6P9a genotypes.** Expression level of AAPP and some members of D7 family genes in CYP6P9a-RR and CYP6P9a-RS mosquitoes in comparison with *CYP6P9a* susceptible mosquitoes. The normalized relative expression of each gene against two housekeeping genes (RSP7 and Actin) is represented on the vertical axis. Letters a, b, c, d, e indicates the absence of significant difference in the expression level of each gene between the three types of mosquitoes.

## Discussion

Recently, mutations in the *GST* epsilon 2 and in the promoter region of the cytochrome P450 *CYP6P9a*, were described as robust molecular markers for tracking metabolic resistance in pyrethroids resistant populations of *An. funestus* [24, 25]. Using these two key markers, this study assess the possible association of GST- and P450-based metabolic resistance to pyrethroids on the feeding process and blood meal volume of *An. funestus*.

### Association of metabolic resistance on blood feeding success

The present study revealed that *CYP6P9a* but not the *L119F-GSTe2* mutation could impact the blood feeding success of *An. funestus* mosquito as possessing the *CYP6P9a* resistant allele increased the likelihood of being successful in blood-feeding. Such selective advantage of *CYP6P9a* resistance allele was also previously reported in a semi-field study in experimental hut trial which observed that homozygous CYP6P9a-RR mosquitoes were significantly more likely to blood feed than susceptible SS [25]. This result suggests that CYP6P9a -mediated metabolic resistance might influence the ability of *An. funestus* mosquito to blood feed. In contrast, the absence of association observed here for the *L119F-GSTe2* mutation needed to be confirmed by further studies as the low sample of L119F-RR homozygous resistance mosquitoes might have biased our analysis. This low number of L119F-RR mosquitoes could itself be linked to unsuccessfully genotyping of this marker in approximately 20% of samples analysed in this study. This important point highlights the need for further studies to improve the optimization of the protocol used in this study for the genotyping of the *L119F-GSTe2* mutation. On the other hands, the mechanism whereby *CYP6P9a*-R resistant allele could influence mosquito feeding is unknown and was not investigated in the present study. One hypothesis to explain this association could be related to the motivation of mosquito to blood feed. In fact, it has been reported that some mosquito individuals that emerged with insufficient teneral reserves require an initial blood meal to compensate for insufficient teneral reserves rather than for egg development during their first gonotrophic cycle [34–36]. This phenomenon is mostly observed in smaller female mosquitoes that emerge with insufficient reserve [2]. Thus, we can presume that *CYP6P9a* resistant mosquitoes which were found significantly smaller than susceptible in the present study were more motivated to blood feed as they were probably the ones requiring more to compensate for their insufficient teneral reserves. However, it's important to note that it was surprising and unusual to observed CYP6Pa-SS mosquitoes bigger that CYP6Pa-RR ones since previous studies often reported larger mosquitoes tend to be more tolerant of insecticides, or that resistant phenotypes are associated with larger body sizes [37, 38]. This unusual observation could be explained by the fact that, instead of using dead dried and unfed mosquito as usually done, in the present study the weight was estimated using alive fresh mosquitoes which were fed with sugar solution until 24 hours before being weighted. With this approach, mosquito's body weight may have been influenced by water and/or elements of sugar digestion that are eliminated when the mosquito is dried. Also, although mosquitoes in the present study were reared in the same tanks (at the larval stage) or cages (in adult stage), pooled regardless their genotype, some important parameters such as larval density, amount of food were not strictly controlled. So we could not exclude that this absence of controlled rearing conditions have significantly influenced mosquito growing and consequently its bodyweight. Overall, it is important to note that the unusual body weight of *An. funestus* mosquito observed here represents one important limitation of our study. This point highlight the need to perform further studies working for instance with dried mosquitoes before confirm our hypothesis about the association of teneral reserve and CYP6P9a mutation. One other approach could be to carry out calorimetric assays comparing teneral reserve

between CYP6Pa-SS and CYP6Pa-RR mosquitoes. We could also simply measure and compare the size of the wings between CYP6Pa-SS and CYP6Pa-RR mosquitoes. These further studies would certainly be more informative on the influence of metabolic resistance on the motivation of *An. funestus* mosquito to blood feed.

### Influence of metabolic resistance on probing time and feeding duration

The influence of metabolic resistance on probing time and feeding duration was assessed in the present study only for *L119F-GSTe2* mutation. Results revealed no significant association of this metabolic resistance gene on the time spent by a mosquito to probe. The absence of association of insecticide resistance on mosquito probing time was also reported for the *knock-down* (*kdr*) resistance gene in *Anopheles gambiae* with no difference in the probing time noticed between genotypes (RR, RS and SS) after exposure to untreated and insecticide-treated net [11]. This seems to indicate that insecticide resistance might not impact the probing duration of *Anopheles* mosquito during blood feeding. However, this hypothesis must be taken with caution as, to our knowledge, and the exception of the present study as well as the one of Diop et *al*, very little information is available on the impact of insecticide resistance on the probing time during mosquito blood-feeding. In the other hand, even if the difference was not statistically significant, mosquitoes possessing an *119F-GSTe2* resistant allele (both homozygous and heterozygous) spent less time taking their blood meal than susceptible. This corroborates with observation previously made for *kdr* mutation in *An. gambiae* with lower feeding duration for homozygous resistant mosquitoes than heterozygote and homozygous susceptible [11]. The non-significant result observed may be due to the low number of resistant mosquitoes in the present study. However, from the results, it could be hypothesized that *L119F-GSTe2* mutation might confer an advantage to homozygous resistant mosquitoes as it was previously reported that rapid feeding reduces the risk to be killed by the host defensive behaviour [11, 39].

### Effect of metabolic resistance on blood meal volume

In this study, we observed that the volume of blood ingested by a mosquito during a single blood feeding was associated with the genotype of the P450 *CYP6P9a* but not with the *L119F-GSTe2*-based metabolic resistance. This suggests that mechanisms involved in metabolic resistance to pyrethroids in *An. funestus* might influence mosquito life-traits differently. However, as already discussed above, we cannot exclude that the absence of the influence observed for *L119F-GSTe2* gene might also be related to the low number of L119F- RR mosquitoes used in the present study. This latter hypothesis seems moreover reinforced by the results of previous studies showing *L119F-GSTe2* mutation [22] and *CYP6P9a*-R resistance gene [40] influencing *An. funestus* fecundity in the same way. The positive association between *CYP6P9a*-R resistant allele and the volume of blood meal is in line with the work of Okoye and collaborators reporting that pyrethroid resistance mechanism in southern African *An. funestus* cause no reduction in fitness of this mosquito [41]. Thus, our finding suggests that the over-expression of *CYP6P9a* gene might probably not compromise the volume of blood ingested of individual mosquitoes carrying the *CYP6P9a*-R resistant allele. Given that activity of P450 monoxygenases as well as blood meal digestion, have been reported to generate an excess production of reactive oxygen species (ROS) increasing oxidative stress which could induce several damages in the mosquito's system that can result to death [42, 43], it could have been expected to see *CYP6P9a*-RR mosquitoes taking lower blood to reduce negative effects of oxidative stress. This observation could certainly be explained by the ability of *Anopheles* mosquitoes to cope with oxidative damage after blood feeding by increasing the antioxidant activity

enzymes including, Cu Zn and Mn superoxide dismutase (SOD), catalase, glutathione peroxides and thioredoxin reductase [44]. This suggests that association between the *CYP6P9a*-R resistant allele and mosquito's blood meal size could be an indirect consequence of some other physiological activities. For instance, because *CYP6P9a* resistant mosquitoes were significantly smaller than their susceptible counterparts, and noting that it has been demonstrated that the amount of teneral reserves is proportional to the body size of mosquito [2], we can presume that the high blood meal volume ingested by *CYP6P9a*-RR mosquitoes might be as a result of a need to compensate for the limited teneral reserves post emergence. In this case, the association observed here could be an indirect consequence of the negative association of *CYP6P9a*-R resistant allele recently observed on the larval development of *An. funestus* [40] resulting to a small body size, and by consequence to insufficient teneral reserves for resistant mosquitoes. Indeed, it was demonstrated that encountering a nutritional environment by *Anopheles* larvae strongly influences adult fitness-related traits such as body size and teneral metabolic reserves [2, 31, 45]. However, our finding did not corroborate with the positive association previously reported between the volume of ingested blood meal and mosquito body size [2]. Further studies will help elucidate the underlying reason of this correlation between *CYP6P9a* genotypes and blood meal size.

## Possible association *of CYP6P9a-R* resistant allele on salivary gland genes expression

To obtain a successful blood meal, a female mosquito must balance the risk of death caused by host defensive behavior against the benefits to feed on a host species that maximize fertility [46]. Salivary components permit mosquitoes to reduce their engorgement time and increase their likelihood of survival [5]. In the present study, we assessed the level of expression of genes encoding for some salivary proteins known to be involved on blood intake process of mosquitoes such as AAPP and members of D7 family proteins [6, 47, 48]. The comparative analysis of the expression level of these genes between *CYP6P9a* genotypes showed no significant difference between mosquitoes bearing the resistant allele and those with the susceptible one. This result suggests that the expression of AAPP and D7 family salivary genes are not associated with the *CYP6P9a* mutation. This observation is intriguing as some salivary genes such as D7 family genes were previously reported to be over-expressed in resistant *An. funestus* mosquito compared to susceptible strain [24, 49–52]. The lack of significance observed with the differential expression of genes in the present study could be explained by the fact that our analyses in this study were performed on mosquitoes obtained after crosses between two different strains and therefore sharing the same background, while other studies compared insecticide resistant field /laboratory mosquitoes and susceptible laboratory strains with different genetic background [24, 49, 52]. The absence of influence of the *CYP6P9a* gene on the expression level of salivary gland genes involved in the blood-feeding process observed in the present study appears to indicate that the association found between this gene and the size of blood meal taken by *An. funestus* mosquito might not be related to the expression of these salivary genes encoding proteins which mediate the blood meal process.

This study revealed that GSTe2-mediated resistance is not associated with the blood meal intake of *An. funestus* mosquitoes, whereas *CYP6P9a*-based resistance to pyrethroids is associated with a feeding success and a higher blood meal size. However, this influence on *Anopheles funestus* blood meal intake is not associated with differential expression of major salivary gland proteins involved in blood-feeding. Given the rapid growth of insecticide resistance, it would be interesting to study how this association could affect the fecundity and the vectorial capacity of *An. funestus* mosquitoes.

## Supporting information

**S1 Table. List of primers used for qPCR analyses.**
(DOCX)

## Acknowledgments

We thank Dr Micheal KUSIMO and Mr Francis NKEMNGO; native speakers of English language for accepting to review and edit this paper.

## Author Contributions

**Conceptualization:** Lynda Nouage, Emmanuel Elanga-Ndille, Charles S. Wondji.

**Data curation:** Lynda Nouage, Emmanuel Elanga-Ndille, Tatiane Atsatse.

**Formal analysis:** Lynda Nouage, Emmanuel Elanga-Ndille, Cyrille Ndo.

**Funding acquisition:** Emmanuel Elanga-Ndille, Charles S. Wondji.

**Investigation:** Lynda Nouage, Emmanuel Elanga-Ndille, Achille Binyang, Magellan Tchouakui, Tatiane Atsatse.

**Methodology:** Lynda Nouage, Emmanuel Elanga-Ndille, Achille Binyang, Magellan Tchouakui, Tatiane Atsatse, Cyrille Ndo, Sévilor Kekeunou, Charles S. Wondji.

**Project administration:** Emmanuel Elanga-Ndille.

**Supervision:** Emmanuel Elanga-Ndille, Sévilor Kekeunou, Charles S. Wondji.

**Validation:** Emmanuel Elanga-Ndille, Cyrille Ndo.

**Visualization:** Achille Binyang.

**Writing – original draft:** Lynda Nouage, Emmanuel Elanga-Ndille.

**Writing – review & editing:** Emmanuel Elanga-Ndille, Achille Binyang, Magellan Tchouakui, Cyrille Ndo, Sévilor Kekeunou, Charles S. Wondji.

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
