## [Decision Letter · Decision Letter 0]

27 May 2020

PONE-D-20-07208

Influence of GST- and P450-based metabolic resistance to pyrethroids on blood feeding in the major African malaria vector Anopheles funestus

PLOS ONE

Dear Dr. Nouage,

Thank you for submitting your manuscript to PLOS ONE. After careful consideration, we feel that it has merit but does not fully meet PLOS ONE’s publication criteria as it currently stands. Therefore, we invite you to submit a revised version of the manuscript that addresses the points raised during the review process.

We look forward to receiving your revised manuscript.

Kind regards,

Basil Brooke, PhD

Academic Editor

PLOS ONE

Journal Requirements:

https://journals.plos.org/plosone/article?id=10.1371%2Fjournal.pone.0103816

https://www.mdpi.com/2075-4450/10/9/265/htm

In your revision ensure you cite all your sources (including your own works), and quote or rephrase any duplicated text outside the methods section. Further consideration is dependent on these concerns being addressed.

3. In your Methods, please describe exactly how volunteers were recruited to provide blood meals in your study.

'This study was funded by a Wellcome Trust Training fellowship (109930/Z/15/Z) awarded to ELANGA N’DILLE Emmanuel.

'The funders had no role in study design, data collection and analysis, decision to

publish, or preparation of the manuscript'

Additional Editor Comments (if provided):

Reviewers' comments:

Reviewer's Responses to Questions

**Comments to the Author**

1. Is the manuscript technically sound, and do the data support the conclusions?

Reviewer #1: Yes

Reviewer #2: Partly

2. Has the statistical analysis been performed appropriately and rigorously? 

Reviewer #1: Yes

Reviewer #2: Yes

3. Have the authors made all data underlying the findings in their manuscript fully available?

Reviewer #1: Yes

Reviewer #2: Yes

4. Is the manuscript presented in an intelligible fashion and written in standard English?

Reviewer #1: Yes

Reviewer #2: Yes

5. Review Comments to the Author

Reviewer #1: It is a technically well executed study and the manuscript well written and presented.

The subject is interesting. I would request that the authors please consider a few points in their revised version:

1. In general, I think “impact” of CYP6P9a-R genotypes on several blood feeding behavior parametres is too strong. Please consider replacing impact (causal factor?) with “possible association” or similar alternative term

For example, Line 358: “Due to the association observed between the CYP6P9a-R genotypes and blood feeding, an attempt was made to assess whether the genotypes of this gene could impact the expression profile of key salivary genes” and 451 influence of CYP6P9a-R resistant allele on salivary gland genes expression

I am not sure what is meant, how would the CYP6P9a-R IMPACT or Influence the expression of salivary glands? (please consider alternative “possible association” term ..)

2. Assuming volume of blood meal ingested by CYP6P9a-R mosquitoes was (sligthly) lower than that of CYP6P9a-S, does that mean vectorial capacity is also affected? Please justify the link, or tone down the statement

Line 45 “This study suggests that P450-based metabolic resistance may increase the blood feeding ability of malaria vectors and potentially impacting their vectorial capacity” contradicts with line

477 “interesting to study how this association could impact the fecundity and the vectorial capacity of An. funestus mosquitoes”

3. The data presented in Figure 4 are not clear what it is, and the variation shown (SEM? SD??) is not normal (or not well analysed and explained). If true, alternative ways could be considered to present this data.

Reviewer #2: I have made several minor comments on the PDF. I have given my comments as an attached word document. The primary issue that I have is that I have not been fully convinced by the arguments, it must be strengthened and there are some rewriting required to make the manuscript easier to read.

6. PLOS authors have the option to publish the peer review history of their article (what does this mean?). If published, this will include your full peer review and any attached files.

Reviewer #1: No

Reviewer #2: No

---

## [Author Response · Author response to Decision Letter 0]

8 Jul 2020

Editor’s comments

1. Please ensure that your manuscript meets PLOS ONE's style requirements, including those for file naming. The PLOS ONE style templates can be found at https://journals.plos.org/plosone/s/file?id=wjVg/PLOSOne_formatting_sample_main_body.pdf and https://journals.plos.org/plosone/s/file?id=ba62/PLOSOne_formatting_sample_title_authors_affiliations.pdf.

Thank you for this remark, the manuscript have been check completely for any formatting mistake.

https://journals.plos.org/plosone/article?id=10.1371%2Fjournal.pone.0103816

https://www.mdpi.com/2075-4450/10/9/265/htm

Answer: This observation was taken into account and we insured to strongly eliminated the overlapping even if we didn’t know exactly which part of the text is overlapping

In your revision ensure you cite all your sources (including your own works), and quote or rephrase any duplicated text outside the methods section. Further consideration is dependent on these concerns being addressed.

Answer: Thank you so much for highlighting this. We have rephrased all the duplicated text detected and referenced the remaining parts as you can see in the revised version

3. In your Methods, please describe exactly how volunteers were recruited to provide blood meals in your study.

Answer: Thank you for this observation. This point was taken into account and a sentence describing briefly how volunteers were recruited to provide blood meal has been added in the revised manuscript (see lines 146 To 149).

Answer: Thank you so much for mentioning this. I created an ORCID iD which allowed to submit the revised version of the manuscript

'This study was funded by a Wellcome Trust Training fellowship (109930/Z/15/Z) awarded to ELANGA N’DILLE Emmanuel.

'The funders had no role in study design, data collection and analysis, decision to

publish, or preparation of the manuscript'

Answer: This observation was taking in consideration and the funding-related text have been removed from the manuscript.

Reviewer #1: It is a technically well executed study and the manuscript well written and presented.

The subject is interesting. I would request that the authors please consider a few points in their revised version:

1. In general, I think “impact” of CYP6P9a-R genotypes on several blood feeding behavior parametres is too strong. Please consider replacing impact (causal factor?) with “possible association” or similar alternative term

For example, Line 358: “Due to the association observed between the CYP6P9a-R genotypes and blood feeding, an attempt was made to assess whether the genotypes of this gene could impact the expression profile of key salivary genes” and 451 influence of CYP6P9a-R resistant allele on salivary gland genes expression

I am not sure what is meant, how would the CYP6P9a-R IMPACT or Influence the expression of salivary glands? (Please consider alternative “possible association” term.)

Answer: We thank the reviewer for this comment. We agree with him that talking about ‘impact’’ in our study could be too strong to characterise our observations. According to its recommendation, we have replaced the term ‘’impact’’ by ‘’ possible association or effect “in the entire revised manuscript.

2. Assuming volume of blood meal ingested by CYP6P9a-R mosquitoes was (sligthly) lower than that of CYP6P9a-S, does that mean vectorial capacity is also affected? Please justify the link, or tone down the statement

Line 45 “This study suggests that P450-based metabolic resistance may increase the blood feeding ability of malaria vectors and potentially impacting their vectorial capacity” contradicts with line 477 “interesting to study how this association could impact the fecundity and the vectorial capacity of An. funestus mosquitoes”

Answer: We thank the reviewer for this remark. Does assumption that volume of blood meal ingested by CYP6P9a-R mosquitoes was (slightly) lower than that of CYP6P9a-S means that vectorial capacity is also affected? That is an interesting question to be investigated. At this stage, since we did not evaluate this aspect, we are just making hypothesis that association between the CYP6P9a-R allele and the volume of blood meal ingested may potentially influence the vectorial capacity. Indeed, since the vectorial capacity is a concept analogous which is a function of the vector's density in relation to its vertebrate host, it could be influenced by the variations in mosquito’s density in one area. So, given that mosquitoes’density hardly depends to mosquito fecundity and that the latter is itself significantly associated with the volume of blood meal ingested, we are hypothesising that, lower volume of blood meal ingested by CYP6P9a-R mosquitoes could affect its fecundity by reducing the number of eggs they lay. The reduction of the number of eggs laid would led to the decrease of the density of CYP6P9a-R mosquitoes in a given population and therefore to a reduction of vectorial capacity of these mosquitoes. However, since we did not assess this impact on the vectorial capacity and because vectorial capacity depends also to other parameters such as infectiousness, the longevity and the behaviour our statement remains one hypothesis and that is why in line 477, we are saying that it would be interesting to investigate this hypothesis through further studies. However, to avoid any confusion for the understanding of our statement, we rewrite the sentence at line 45 in the revised manuscript as followed: “This study suggests that P450-based metabolic resistance may influence the blood feeding process of Anopheles funestus mosquito and by consequence its ability to transmit malaria vectors and parasites.”

3. The data presented in Figure 4 are not clear what it is, and the variation shown (SEM? SD??) is not normal (or not well analysed and explained). If true, alternative ways could be considered to present this data.

Answer: We thanks the reviewer for this comment and we are totally agree with him that there is sometime wrong with this graphic. Indeed, since fold changes are not normally distributed data, there is no reason to represent them with standard errors. Moreover, these standards errors were not well estimated. After reanalysing these data we found that relative expression of each gene is the great way to present this result. Thus, since the trend of the result is the same, in the revised manuscript we replace the former figure 4 by a new one presenting the comparative relative expression of each between CYP6P9a-RR, CYP6P9a-RS and CYP6P9a-SS mosquitoes. Also, some few modifications were made in the part entitled “Expression profile of AAPP and D7 family salivary genes according to CYP6P9a-R genotypes” of the results section of the revised manuscript.

Reviewer #2:

Although the authors have done work that is reasonably methodologically sound, I am not fully convinced by the conclusions, rather than having a problem with the findings. The key problem is that the argument of the association of the genotype with behaviours is not strong enough. I will outline some of the key issues that need to be addressed to strengthen the argument enough to be published. The experiments are largely sound, but there needs to be a large scale restructuring of the discussion and the conclusions drawn, as the arguments are not strongly convincing.

Major comments:

There is a substantial body of work about the effects of metabolic resistance and life history. The novel factor of your work is that your work involved wild specimens, where previous studies on metabolic resistance and its effects were on laboratory strains where the resistance mechanisms were characterised. Although the metabolic screening tools are useful, this does not mean that studies on metabolic resistance could not happen before it. The success rate of genotyping of the wild specimens (78% success rate) is concerning. The substantial amount of unsuccessfully genotyped individuals could have altered the findings. This must be acknowledged.

Answer: We agree with the reviewer that our study is not the first one on metabolic resistance since some other studies were already done. However our study is among the most recent ones using a DNA based marker to investigate the physiological impact of metabolic resistance in Anopheles mosquitoes. Concerning the L119F-GSTe2 genotyping, we agree with the reviewer that the number of unsuccessfully genotyped individuals is not negligible and could have altered our findings for this marker. This aspect was taken into account as it could be the source of low number of L119F-GSTe2-RR mosquitoes in this study. That is why in the first manuscript we have indicated that our findings for the L119F-GSTe2 mutation could have been biased by the low number of L119F-GSTe2-RR mosquitoes. However to be more understandable we added more comments in the revised manuscript (see lines 410 to 414). Nevertheless, since our analyses were performed on approximately 78% (360/457) of our total sample we think that this sample size was more than enough for this assessment. 

The authors need to guide the readers a little more. There are times in the methods section where it is confusing as to why the methods were being employed, but this was clarified in the results section. It would make the MS easier to read if the reason for the methodology was introduced prior to the description of the method.

Answer: We thanks the reviewer for this suggestion. This remark have been taken into consideration and the methods section were rewritten in the revised manuscript to help the readers understanding why each method was employed in the study.

During the PCR methods section, it seems that the different sections were written by different people. In some parts it is satisfactory, and in others, it seems that the methods come from a dissertation. PCR methods in a manuscript include concentrations and not volumes, and these sections are more suited for a standard operating procedure than a manuscript. Centrifugation data must be reported in rcf (g) not rpm. Please ensure that the language of the molecular methods is appropriate.

Answer: All these remarks were taken into consideration. We have revised the PCR section in the revised manuscript and we replaced the volume of the reagents by the concentrations. We also converted the centrifugation data from rpm to rcf.

The weight of the An. funestus mosquitoes you describe is very high. In my experience mosquitoes weighing close to a milligram normally have wingspans of between 3.5 to 4mm, and these are very large An. gambiae. This result is therefore quite startling. It also leads to questions about the conclusions about genotype and size.

Answer: The observation of the review about the size of the weight of mosquitoes used in this study is understandable, since these mosquitoes look bigger than what usually reported. However, we think that there are important points to consider here before comparing the weight of our mosquitoes and those from previous studies. These points could certainly explained the huge differences between our study and other ones. For instance, instead of weighing dead and dried mosquitoes as commonly done in previous studies, mosquitoes used in our study were not killed and dried before being weighted. So mosquitoes’ weight in our study is not a dried weight. This difference in the method used is important since there are numerous components that can influence mosquito’s weight which are eliminated when mosquito is drying. Additionally, in our study we worked with mosquitoes of 3 to 7 days old whereas other previous studies mainly worked with mosquitoes of 1 day old maximum. This difference in age can also lead to the difference of the weight between two mosquitoes. One other important element that could explain the bigger weight of mosquitoes used in this study compared to other ones, is the fact that unlike of what done in previous studies, mosquitoes here were fed with sugar solution before being weighting after being starved during 24h. Overall, all these parameter could explained why our mosquitoes look bigger than usual. Furthermore, is important to noted that the weight our mosquitoes is not so different to what Roitberg and its colleagues obtained by weighting also fresh mosquitoes (Roitberg BD, Mondor EB, Tyerman JGA (2003) Pouncing spider, flying mosquito: blood acquisition increases predation risk in mosquitoes. Behavioral Ecology, Vol. 14 No. 5: 736–740, DOI: 10.1093/beheco/arg055). However, it could be interesting in further studies to assess the weight of Anopheles funestus mosquito when he is dried. 

Furthermore, it is also concerning that you talk about size, and the relationship between genotype and size, without describing how you controlled for the larval diet. If this was not controlled for, then genotype may be related to larval feeding propensity that resulted in larger adults, which is a slightly different situation. It may also simply be that there was feeding variation between cohorts. Therefore, if you cannot describe how larval feeding was standardised, then conclusions about size cannot be made.

Answer: We completely agree with the reviewer that the mosquito’s weight hardly depend of larval rearing and feeding condition. Larval diet strongly impact the weight of mosquitoes. But in the case of our study, the potential bias due to the larval rearing condition could not impact our findings. Indeed, at the larval stage, mosquitoes were reared and fed in the same conditions since eggs laid by parent were put in the same trays regardless the genotypes. It’s only at the adult stage and after the weighting and the blood-feeding that mosquitoes were genotyped and separated according to their genotypes. Before the genotyping mosquitoes at all stages were pooled and reared in the same conditions. This is one of the main advantage of using the molecular markers for such assessment since they help to reduce all the confounding factors which could impact the outcome of the study.

Something that also jumps out is that CYP6P9-SS adults are bigger. This is quite unusual, as larger mosquitoes tend to be more tolerant of insecticides, or that resistant phenotypes are associated with larger body sizes (eg: Osuwu et al: Sci Rep. 2017 Jun 16;7(1):3667, Jeanrenaud et al, PLoS One. 2019 Apr 18;14(4):e0215552). Such an unusual result must be discussed.

Answer: Thanks for this remark. We were also surprised by this result since we were expecting to observe the opposite pattern according to the fact that resistant phenotypes are often reported associated with larger body sizes. As suggested by the reviewer, this point is now discussed in the revised manuscript (See lines: 422-437).

Line 375-396: Impact of metabolic resistance on blood feeding success

This particular argument does not hold water. Firstly, you have just given the definition of an anautogenous mosquito. Secondly, you hypothesize that CYP6P9 could affect feeding because of teneral reserves. You have not examined this, and crucially, as have described above, you do not describe how feeding was controlled for and this is crucial for comments on size. Also, as I have described above, the mosquitoes you have described are extremely large for An. funestus, so I don’t think poor teneral reserves can account for the findings you have described. Related to this, later there are discussions about resistance and blood feeding, hypothesising that the smaller resistant mosquito (which is unusual in itself) needs to take more blood because they have lower teneral reserves. This line of argument is not well supported as there was no evidence that food quantity was controlled, and therefore there cannot be arguments about teneral reserves. Teneral reserves are not only represented by size, and if there is an argument to made about this, there are simple calorimetric assays described in the MR4 manual to measure this. Finally, the fact that these mosquitoes seem to be very large for their species, it seems problematic to ascribe this behaviour to poor teneral reserves. 

Answer: We well understand the concern highlighted by the reviewer. However, as we mentioned above, we have eliminated all potential bias related to feeding and rearing conditions since our mosquitoes were genotyped and sorted according their genotype only after blood-feeding experiments mosquitoes used in this study were reared in the same conditions. Indeed, larvae were pooled in the same tray, fed in the same conditions. After emergence, adult mosquitoes were in the same cages and were fed with the same sugar solution. It is only after all the experiments and genotyping that the link between body size and the genotype was assessed. However, although we have reduced the bias related to breeding conditions, it cannot be excluded that the weight of mosquitoes was influenced by the digestion of the sugar solution which could have been incomplete in some individuals. Other studies based on calorimetric analyses could probably allow a better comparison of the quantity of teneral reserves between resistant and susceptible mosquitoes. This aspect was pointed out and discussed in the revised manuscript.

 Line 398-417 Impact of metabolic resistance on probing time and feeding duration

Besides the comment that I have made on the document that it is not worth discussing results that are not statistically significant, there is another methodical flaw that is revealed in this discussion. By the authors’ admission, there were not a large amount of resistant heterozygotes. Although crosses were performed to generate heterozygotes, if Fumoz was available for crossing, surely uncrossed Fumoz would have been a useful source of RR mosquitoes? Is there a reason why RR and SS individuals were not obtained for study from the original strains?

Answer: We thank the reviewer for the comment. Since our study aimed to assess the impact of metabolic resistance in important feeding parameters of Anopheles funestus mosquito, it was crucial to work with mosquitoes shearing the same genetic background. For this purpose we worked with Fumoz (full resistance) and FANG (full susceptible) strains. Since the genetic background of these two lab strain is huge different, we thought that it would not have been reasonable to work separately with RR (from Furmoz) and SS (from FANG). So to limit the impact of this genetic background differences the option chosen and which is recommended (Thiago Affonso Belinato and Ademir Jesus Martins (March 2nd 2016). Insecticide Resistance and Fitness Cost, Insecticides Resistance, Stanislav Trdan, IntechOpen, DOI: 10.5772/61826. Available from: https://www.intechopen.com/books/insecticides-resistance/insecticide-resistance-and-fitness-cost), was to work with mosquitoes generated after crossing FANG and Fumoz. Working with individuals generated from crosses avoid our funding being influenced by other several genetic difference.

Line 418-450: Impact of metabolic resistance on blood meal volume

There are some issues in this section.

Line 427-431. The authors describe that “the positive association CYP6P9a-R resistant allele and the volume of blood meal is a bit surprising knowing that activity of P450 monoxygenases as well as blood meal digestion, have been reported to generate an excess production of reactive oxygen species (ROS) increasing oxidative stress which could induce several damages in the mosquito’s system that can result to death [38,39]. This is not really that surprising if you consider that the resistant An. funestus strains Fumoz and in particular Fumoz R has a marked capacity to cope with oxidative stress, both phenotypically and due to increased gluthatione peroxidase and catalase activity (Oliver and Brooke PLoS One. 2016 Mar 10; 11(3):e0151049): Your reference cites information about oxidative stress in Aedes. There are better references, as there is work on oxidative stress in Anopheles, not only the previously mentioned which shows that oxidative stress in An. funestus specifically is quite different to Aedes. Champion and Xu (Sci Rep 8, 13054 (2018)) show the complexity of the interaction between oxidative stress, insecticide resistance and fecundity in An. gambiae. Crucially, what they show a relationship between oxidative stress and resistance, with resistant An. gambiae also has a greater oxidative stress capacity. Over and above this, epsilon class GSTs are associated with oxidative stress defence. In the selections, did these genotypes segregate separately (was this checked), because it will be difficult to discuss oxidative stress in relation to CYP6P9 (where P450s do increase oxidative stress) if the effect of the GSTe2 in the same individual has not been considered. 

Answer: The point highlighted by the reviewer is very interesting and was taken into consideration. We really found interesting all papers mentioned by the reviewer, since they help us improving our discussion section in the revised manuscript. Nevertheless, We would like to precise that the effect of GSTe2 and P450s where not assessed on the same mosquitoes since the GSTe2, unlike to the duplicated CYP6P9a and b, is not implicated in resistance in FUMOZ strain.That is the reason why we used two different strains to assess the impact of metabolic resistance in An. funestus blood-feeding process: i) FUMOZ lab strain to assess the impact of CYP6P9a and ii) for the impact of GSTe2 we used the field collected mosquitoes from Cameroon where this gene where previously shown to be overexpressed in DDT and Permethrin resistant mosquitoes (Riveron JM, Yunta C, Ibrahim SS, Djouaka R, Irving H, et al. Genome Biology (2014)). Furthermore, we agree with the reviewer that it will be difficult to discuss oxidative stress in relation to CYP6P9 if the effect of the GSTe2 in the same individual has not been considered, since both enzyme have opposite effect. In our study we did not assess this point because as said above, among the two genes concerned in our study, only the CYP6P9a is reported over-expressed and evolved in insecticide resistance of FUMOZ-R strain. However, although GSTe2 is not over-expressed and evolved in insecticide resistance in Fumoz strain, I would have been interesting to measure its activity in CYP6P9a-RR mosquitoes before and after blood-feeding. This would have been more informative about the mechanism used by the CYP6P9a-RR mosquito to cope with the oxidative stress generated by the positive association between high blood meal volume and the CYP6P9a-R mutation. In the same vein, the assessment of the activity of other antioxidant agents would be helpful for the great understanding of the control of oxidative damage by CYP6P9a-RR mosquito after blood-feeding. All these points are now included in the discussion section in the revised manuscript.

Another major issue that I have with this argument has to do with the argument about fitness costs in An. funestus. Although your reference for this is a manuscript where fitness costs are noted, this manuscript examined it in crosses, as you have done in this experiment. However, that study and the present study ignore the work of Okoye et al (Bull Entomol Res. 2007 Dec; 97(6):599-605), which did not find a fitness cost to resistance in Fumoz. Their conclusion is supported by the fact that Fumoz-R maintains high deltamethrin resistance intensity without selection. Therefore, a fitness cost cannot be unambiguously assigned to the resistant funestus.

Answer: First of all, we would like to reassure the reviewer that our study did not ignore the work of Okoye et al. We did not referred to this previous work because the authors used two strains of different genetic background to evaluate the fitness associated with resistance in Fumoz since they do not have a DNA-based marker. The main approach for such assessment is to cross the resistant and the susceptible strains so that the fitness cost will be evaluated on mosquitoes with the same genetic background. This is the approach used in this study and previously by Tchouakui et al 2020. However, since our findings appeared to show no reduction of the fitness of CYP6P9a-RR mosquitoes as previously observed by Okoye et al in phenotypically resistant FUMOZ strain, we have revised and restructured our discussion. In the revised manuscript we are now taking into account the fact that our findings seem to suggest that the CYP6P9a-based metabolic resistance might not probably compromise some life straits of Anopheles funestus mosquito, including its ability to have a blood meal and the volume of blood it is able to ingest. Because of these modification of the discussion of our results, the paper of Okoye et al, is now cited in the revised manuscript. 

The studies on salivary gene expression are problematic. I assumed that there would not be a different as the error bars (if they are error bars and not standard deviation) are huge. This is concerning and this is not a good quality result for publication. Is there an explanation for this?

Answer: We thanks the reviewer for this comment. As already said above, there was sometime wrong with this graphic. Indeed, since fold changes are not normally distributed data, there is no reason to represent them with standard errors. Moreover, these standards errors were not well estimated. After reanalysing these data we found that relative expression of each gene is the great way to present this result. Thus, since the trend of the result is the same, in the revised manuscript we replace the former figure 4 by a new one presenting the comparative relative expression of each between CYP6P9a-RR, CYP6P9a-RS and CYP6P9a-SS mosquitoes. Also, some few modifications were made in the part entitled “Expression profile of AAPP and D7 family salivary genes according to CYP6P9a-R genotypes” of the results section of the revised manuscript

Minor comments

Throughout the document: change homo or heterozygote to heterozygous where it is used as an adjective rather than a noun.

Thanks to the reviewer, homo or heterozygote has been changed to heterozygous thought out the manuscript

There are numerous small comments annotated on the manuscript that must be addressed.

Thanks a lot for all the attention on the manuscript all the comments have been addressed as you can see in the revised version

How was RNA quality assessed?

Answer: This was mainly evaluated based on the ratio obtained from Nanodrop with ratio ranged from 1.8 - 2.0. Using the bioanalyzer could have been the best option for that, but; unfortunately this expensive equipment is not yet available in our lab.

I do not think that actin was the best choice of housekeeping gene for a study about a blood feeding response. Actin would not stay stable during blood feeding as the abdominal expansion associated with blood feeding would alter actin expression.

Answer: Actin has been used for many studies on the expression profiling of insecticide resistance genes in An. funestus. Probably, the abdominal expansion associated with blood feeding can alter actin expression as mentioned the reviewer but in our study we assessed genes expression in the salivary glands. Nevertheless this has to be taken into account for further studies on this aspect.

Line 291: This is quite unusual. The laboratory strains would have fed quite well if they were allowed to feed for 30 minutes. How old were these mosquitoes when they were fed? Over and above the parameters I described before, it may be that the mosquitoes did not feed well, as 3 days is not the optimal feeding age for An. funestus.

Answer: The mosquitoes used for the experiments aged between 3-7 days old. Probably 3 days is not the optimal feeding age for An. funestus but all the mosquitoes did not have 3 days. Also this cannot impact the outcome of the association between the markers and the feeding since the age cannot affect only one of the three genotypes given that mosquitoes were reared, fed and kept together.

Did the mean amount of blood taken in the GST genotype differ significantly from the CYP6 genotype?

Yes a significant difference was found between GST and CYP6, with mosquitoes displaying low mean blood meal volume for GSTs (P<0.0001). However, since our aims was to compare these two mosquitos’ strains, we did not include this analysis in our manuscript. Also, we thought that this analyse might have been biased by the fact that one type of mosquito was lab strain (CYP6P9a-mosquitoes) which is most able to take it blood meal in lab condition than field collected (GSTe2 mosquitoes) field.

Line 368: This is not true. There is a body of work that describes the interplay between metabolic resistance and longevity and stress response, among others. It is not as well examined as target site resistance, but there is a body of work on the subject. If the statement was amended to say that this is so for wild specimens, that is true because the studies were primarily studied in lab strains where the metabolic resistance profile had been defined.

This comment was take into consideration and Line 368 was removed in the revised manuscript

---

## [Decision Letter · Decision Letter 1]

13 Aug 2020

PONE-D-20-07208R1

Influence of GST- and P450-based metabolic resistance to pyrethroids on blood feeding in the major African malaria vector Anopheles funestus

PLOS ONE

Dear Dr. Nouage,

Thank you for submitting your manuscript to PLOS ONE. After careful consideration, we feel that there are a few more adjustments to address as detailed below. Therefore, we invite you to submit a revised version of the manuscript that addresses the points raised during the review process.

We look forward to receiving your revised manuscript.

Kind regards,

Basil Brooke, PhD

Academic Editor

PLOS ONE

Reviewers' comments:

Reviewer's Responses to Questions

**Comments to the Author**

1. If the authors have adequately addressed your comments raised in a previous round of review and you feel that this manuscript is now acceptable for publication, you may indicate that here to bypass the “Comments to the Author” section, enter your conflict of interest statement in the “Confidential to Editor” section, and submit your "Accept" recommendation.

Reviewer #2: (No Response)

2. Is the manuscript technically sound, and do the data support the conclusions?

Reviewer #2: Yes

3. Has the statistical analysis been performed appropriately and rigorously? 

Reviewer #2: Yes

4. Have the authors made all data underlying the findings in their manuscript fully available?

Reviewer #2: Yes

5. Is the manuscript presented in an intelligible fashion and written in standard English?

Reviewer #2: Yes

6. Review Comments to the Author

Reviewer #2: I do appreciate the careful attention to comments and corrections made by the authors. I am generally happy with the efforts that have been made to improve the manuscript, but I do still have a few comments that require attention. These issues can be corrected to the satisfaction of the editor.

There are a few small editorial comments highlighted on the manuscript.

Minor Comments

1. I am still not convinced by the explanation given regarding the size of the mosquitoes. Stating that the excessive size of the mosquitoes is due to fresh mosquitoes being used rather than dried mosquito is not an acceptable explanation. One of the defining features of An. funestus is that these are relatively small mosquitoes, with this being used as a diagnostic feature for morphological identification, with mosquitoes typically being characterized as having a smaller wingspan than 3.5mm. Therefore, having mosquitoes 1mg per mosquito and larger is quite incongruent, as this is the size of a large An. gambiae complex mosquito. The argument is not helped by the authors quoting that their findings are in line with a study that assessed the size of An. gambiae. I understand that there is not much that can be done about this, but this anomaly must be acknowledged.

2. The next issue that has not been satisfactorily answered by the authors is the effect of larval rearing. The authors state that “potential bias due to larval rearing conditions could not impact findings”. This is absolutely not true, as larval rearing conditions would affect the adult blood feeding behaviour, regardless of how you segregate the mosquitoes. Very shortly afterwards, you state that “mosquitoes at all stages were pooled and reared in the same conditions”. This is what needs to be clarified. Did this same conditions include being fed a controlled amount of food? If yes, this must be described, if not it must be acknowledged.

3. Although the authors have satisfactorily explained their motivation for crossing Fumoz and Fang, it is worth mentioning that this particular experimental set up may be flawed. It must be noted that An. funestus ss has two mitochondrial clades, and these have biological differences. Therefore, future studies need to check the clades of the resultant hybrids, and it is worthwhile checking the clades of the wild material. Although this cannot be attended to for this experiment, it is worth noting that when creating these hybrids, that mitochondrial clade hybrids can also be created, and if the authors continue to plan these type of experiments, this is something that must be taken into consideration.

7. PLOS authors have the option to publish the peer review history of their article (what does this mean?). If published, this will include your full peer review and any attached files.

Reviewer #2: No

---

## [Author Response · Author response to Decision Letter 1]

15 Aug 2020

1. I am still not convinced by the explanation given regarding the size of the mosquitoes. Stating that the excessive size of the mosquitoes is due to fresh mosquitoes being used rather than dried mosquito is not an acceptable explanation. One of the defining features of An. funestus is that these are relatively small mosquitoes, with this being used as a diagnostic feature for morphological identification, with mosquitoes typically being characterized as having a smaller wingspan than 3.5mm. Therefore, having mosquitoes 1mg per mosquito and larger is quite incongruent, as this is the size of a large An. gambiae complex mosquito. The argument is not helped by the authors quoting that their findings are in line with a study that assessed the size of An. gambiae. I understand that there is not much that can be done about this, but this anomaly must be acknowledged.

Answer: We thank the reviewer for this remark he highlights once again. We agree with him that the value of the weight of our mosquitoes shows some potential anomalies that might have an important influence on the results of our study. Because there is not much that can be done about in the present study, we have however acknowledged this anomaly of the mosquito’s weight in the revised manuscript (see lines 430-442). If this type of study is repeated in our lab, we will hardly take into account the recommendation of the reviewer.

2. The next issue that has not been satisfactorily answered by the authors is the effect of larval rearing. The authors state that “potential bias due to larval rearing conditions could not impact findings”. This is not true, as larval rearing conditions would affect the adult blood-feeding behaviour, regardless of how you segregate the mosquitoes. Very shortly afterwards, you state that “mosquitoes at all stages were pooled and reared in the same conditions”. This is what needs to be clarified. Did these same conditions include being fed a controlled amount of food? If yes, this must be described, if not it must be acknowledged.

Answer: Even if our mosquitoes were reared and kept together under the same conditions and regardless their genotypes, we can recognize that since we did not strictly control the amount of food given to mosquitoes at larval ad adult stages, some biases could have been generated during the rearing. These potential biases might have to influence the weight of the mosquitoes and subsequently, some results obtained. This limitation of our study is now acknowledged in the revised manuscript (see lines 430-442). For further studies, it will be important to work under controlled condition to manage and reduce biases related to the weight of the mosquito.

3. Although the authors have satisfactorily explained their motivation for crossing Fumoz and Fang, it is worth mentioning that this particular experimental set up may be flawed. It must be noted that An. funestus ss has two mitochondrial clades, and these have biological differences. Therefore, future studies need to check the clades of the resultant hybrids, and it is worthwhile checking the clades of the wild material. Although this cannot be attended to for this experiment, it is worth noting that when creating these hybrids, that mitochondrial clade hybrids can also be created, and if the authors continue to plan these type of experiments, this is something that must be taken into consideration.

Answer: This suggestion of the reviewer is fully pertinent. This is something we did not pay attention to in our study. As the reviewer suggested for further studies crossing FUMOZ and FANG, we should assess the clades of the hybrids we will work with. Furthermore, this observation of the reviewer will allow us checking these clades of the resultant hybrids from crosses we are currently performed in our lab to select both full susceptible and resistant strains from local An. funestus ss population.

---

## [Editor Report · Decision Letter 2]

27 Aug 2020

Influence of GST- and P450-based metabolic resistance to pyrethroids on blood feeding in the major African malaria vector Anopheles funestus

PONE-D-20-07208R2

Dear Dr. Nouage,

We’re pleased to inform you that your manuscript has been judged scientifically suitable for publication and will be formally accepted for publication once it meets all outstanding technical requirements.

Kind regards,

Basil Brooke, PhD

Academic Editor

PLOS ONE
---

## [Editor Report · Acceptance letter]

31 Aug 2020

PONE-D-20-07208R2

Influence of GST- and P450-based metabolic resistance to pyrethroids on blood feeding in the major African malaria vector Anopheles funestus

Dear Dr. Nouage:

I'm pleased to inform you that your manuscript has been deemed suitable for publication in PLOS ONE. Congratulations! Your manuscript is now with our production department.

Kind regards,

on behalf of

Dr Basil Brooke 

Academic Editor

PLOS ONE